# Beyond athletic development: The effects of parkour-based versus conventional neuromuscular exercises in pre-adolescent basketball players

Mark David Williams[1,2]*, Andrew Hammond[3], Jason Moran[1]

1 School of Sport, Rehabilitation and Exercise Sciences, University of Essex, Colchester, Essex, United Kingdom, 2 Department of Sport, Health and Wellbeing, Writtle University College, Chelmsford, Essex, United Kingdom, 3 Independent scholar, Colchester, United Kingdom

* mark.williams@writtle.ac.uk

**Data Availability Statement:** All relevant data are within the paper and its Supporting Information files.

## Abstract

The purpose of this study was to compare the effects of a parkour-based warm-up to a conventional neuromuscular training (NMT) warm-up on the athletic capabilities of youth basketball players. This was examined through two arms: In Investigation 1, the aims were to measure the effects of the two warm-ups on physical measures of athletic performance in prepubescent basketball players. Using post-intervention semi-structured interviews, Investigation 2 aimed to gain insights from the players in relation to the perceived benefits of the two warm-ups. Pre-adolescent children were recruited from two youth level basketball teams. Participants from one club were randomly assigned to either a conventional NMT warm-up group or a parkour warm-up group, while a control group was formed of participants from the second club. Participants of both experimental groups were required to complete a 15-minute warm-up once per week before their regular basketball practice across 8-weeks. For both groups, the coach adopted the same pedagogical approach, utilising a guided discovery strategy. Pre-post test measures of overhead squat performance, countermovement jump, and 10-metre sprint speed were recorded in all three groups. Additionally, pre-post measures were recorded for a timed parkour-based obstacle course for the two experimental groups. No significant between-group differences were found between pre- and post-test measures. However, analysis using Cohen's *d* effect sizes revealed improvements in both intervention groups versus the control. Moreover, between group effect size differences were observed between the two experimental groups. Following the intervention, participants from both experimental groups were also invited to take part in a post-intervention semi-structured interview to discuss their experiences. The thematic analysis of these semi-structured interviews revealed three higher order themes: *Enjoyment; Physical literacy;* and *Docility*; of which the two former themes appear to align to constructs relating to the wider concept of physical literacy. In summary, warm-ups designed to improve athleticism can include less structured and more diverse movement skills than are typical of conventional NMT warm-ups. Specifically, we provide evidence that advocates for warm ups that include parkour-related activities alongside conventional NMT exercises to preserve

**Funding:** The author(s) received no specific funding for this work.

**Competing interests:** The authors have declared that no competing interests exist.

physical fitness qualities and to simultaneously evoke a sense of enjoyment, fun, and purpose. The benefit of such activities may extend beyond athletic development and, more broadly, contribute to the development of physical literacy.

## Introduction

Experts argue that participation in youth sports such as basketball is a healthy activity for youngsters [1,2]. Youth sports (under the right conditions) are effective in developing what Whitehead [3,4] has described as *physical literacy* (PL). Young et al. [5,6] draw on the International Physical Literacy Association to define PL as "the motivation, confidence, physical competence, knowledge and understanding to value and take responsibility for engagement in physical activities for life" (International Physical Literacy Association [IPLA], 2017). Young and colleagues highlight that Whitehead [7] drew on existential and embodied phenomenological theories to define PL "as a holistic concept which focuses on developing the whole person; mind and body as one," (p. 948) [5,6].

Although engagement in youth sports may contribute to the development of PL, early specialisation (which is defined as year-round participation and competition within a single sport [8,9]) may lead to the underdevelopment of 'fundamental movement skills' (FMS) [10–13]. Commonly, FMS include object manipulation, locomotor capabilities and balance, and are considered to be the building blocks for more advanced athletic movements, such as kicking, throwing and striking skills, and advanced techniques within sports [14–17]. Thus the youngster who is engaged in organised sport might not exhibit competency in the FMS that underpin sport specific skills (SSS) [12,18]. Moreover, owing to strong associations between child motor competence in movement skills and levels of self-confidence, where the possession of FMS is considered integral to PL [6,19,20], early specialisation may impede PL development.

Despite the purported issues relating to early specialisation, some researchers argue that these issues are overly simplistic [6,18] and not fully understood [21]. However, in response to the perceived threat of early specialisation, National Governing Bodies (NGBs) have developed numerous NMT programmatic interventions (e.g., the FIFA 11+, Basketball England's Starting 5, and the English Rugby Union's Activate). Typically, NMT programmes comprise a range of FMS, balance, stability, and muscle strengthening exercises to prepare young athletes for the rigours of their sport [22–24]. Furthermore, to encourage their implementation, the aforementioned NMT programmes have been devised to be conveniently integrated within the warm-up to regular sports training, requiring only ∼20 minutes to complete, thus ensuring athlete compliance and making them relatively time-efficient to execute [25,26]. Such programmes have been found to enhance athletic capabilities and to address factors that are associated with injury incidence [22,25,27,28].

In addition to enhancing athletic capabilities in youth athletes, the efficacy of NMT programmes may also relate to the variability of movement patterns presented within such programmes. Accordingly, the performance of varied movement patterns reduces the persistent mechanical stress on the same soft-tissue structures through repeated exposure to SSSs, while concomitantly developing a greater breadth of FMS [29,30]. Indeed, owing to their high levels of neural plasticity–especially during pre-pubescence, youth athletes who are exposed to NMT stimuli may develop motor control more readily [28,31]. Therefore, in place of a rigidly prescribed NMT-programme that might limit the breadth of movement skills developed, less structured forms of movement training (as is often emphasised for youngsters within athletic

development models [17,37,38])may be more effective. One such activity that may inherently provide exposure to a richer breadth of movement skills through its low structured, guided-discovery coaching approach, is parkour [32].

Previously, parkour has been proposed as an activity to develop FMS and athletic capabilities that can be transferred to SSS [32,33]. Indeed, anecdotally, there appears to have been an increase in the amount of strength and conditioning (S&C) coaches using parkour-based concepts with young athletes to develop movement skills. Often, these coaches have cited the importance of activities being less structured than conventional S&C training forms, as well as being engaging for young athletes to participate in. In this regard, typically, parkour adopts a guided discovery approach to learning that is self-paced and enables the participant to explore their capabilities in the absence of strict technical models [32,34]. Of further relevance, recently, significant associations have been identified between performance in the agility T test, standing long jump and countermovement jump, and higher performance in a parkour obstacle course [35]. Accordingly, parkour has been suggested to be an efficacious, yet still unproven, way to develop transferable movement skills for youth athletes [36]. However, to date no research has examined this theory empirically. Therefore, the purpose of this study was to compare the effects of a parkour-based warm-up to a conventional NMT warm-up on athletic performance measures in youth basketball players, implemented using a low-structured, guided discovery coaching approach. This was examined through two arms: In Investigation 1, the aims were to measure the effects of the two warm-ups on physical measures of athletic performance in prepubescent basketball players. We hypothesised that there would be no differences in outcome measures in response to the respective NMT protocols. Due to the novel concept of using parkour-based activities within the warm-up, using post-intervention semi-structured interviews, Investigation 2 aimed to gain insights from the players in relation to the perceived benefits of the two warm-ups.

## Materials and methods

### Participants

A total of 34 youth basketball players (mean age 11.4 ± 0.67 years) consented to participate in the in the pre-post study design across an 8-week intervention period. To increase homogeneity of the population sample [37], participants were recruited using convenience sampling from four (2 boys' teams and 2 girls' teams) youth basketball teams (under 12 years of age between the months of January and December) from two clubs registered and affiliated with the NGB, Basketball England. Participants from one club were randomly assigned to either a Conventional NMT warm-up group or a Parkour warm-up group. To prevent cross-group contamination, the control group was recruited from the second club. For inclusion in the study, all participants were to be classified as pre-peak height velocity (PHV), by upon the prediction equations by Mirwald et al. [38] (<- -0.5 years from PHV), have at least one year's basketball playing experience, and be free from injury. Exclusion criteria were > 0.5 years from PHV, absence from one testing session, and missing three or more training sessions. All experimental procedures and risks were explained fully, both verbally and in writing. The written consent and assent were obtained from the children and their parents/guardians. Ethical approval of the study was granted by the institutional research ethics committee of the authors university and in accordance with the latest version of the Declaration of Helsinki.

### Phase 1 –Quantitative measures and analysis

**Testing procedures.** All testing was carried out by the first author and took place in gymnasiums across two sites used by the respective basketball clubs for regular practice. Testing

took place one week before and after the eight-week intervention period and included: anthropometry (height, seated height, mass), overhead squat (OHS) assessment, countermovement jump (CMJ), 10-m sprint and, for the experimental groups only, a parkour speed-run. To estimate participant maturity status, anthropometric measures were recorded using medical grade digital scales and stadiometer (Seca, Birmingham, United Kingdom) and entered into a sex-specific equation to predict maturity offset [38]:

Girls: Maturity Offset (years) = -9.376 + (0.0001882 x (leg length x sitting height)) + (0.0022 x (age x leg length)) + (0.005841 x (age x sitting height))–(0.002658 x (age x mass)) + (0.07693 x (mass by stature ratio x 100));

and

Boys: Maturity offset (years) = -9.236 + (0.0002708 x (leg length x sitting height)) + (-0.001663 x (age x leg length)) + (0.007216 x (age x sitting height)) + (0.02292 x (mass by stature ratio x 100)).

For the OHS assessment, participants were instructed to hold a wooden dowel with extended arms above the crown of their head and, while maintaining the OH position, squat as low as possible. Following three warm-up trials, three further repetitions were performed and recorded using the motion analysis system, HumanTrak (Vald Performance, Brisbane, Qld, Australia). The sum of knee flexion angle for both limbs for the OHS were averaged for the three repetitions and used in the analysis.

To measure the CMJ, participants were required to jump with their hands placed upon their hips and instructed to descend to a self-selected countermovement depth before immediately jumping as high as possible. Following three warm-up trials, participants performed three test trials on dual portable force platforms (ForceDecks, Vald Performance, Brisbane, Qld, Australia), with at least 20 seconds between trials. The average of the three jumps were analysed.

For acceleration speed, electronic timing gates were used (Brower Timing Systems, Draper, Utah, USA). Following a standardised warm-up comprising submaximal running efforts over a 10-m distance and two practice trials at maximal intensity, each participant completed three trials with at least 60-seconds recovery between trials. Participants began each trial in a *two-point* position 50 cm behind the first timing gate and were then instructed not to countermove ahead of their first step forward, and to sprint through the end timing gate. The average of the three trials was used in the analysis.

The speed-run route was designed in accordance with Strafford et al. [33,35] and in collaboration with an experienced parkour coach and athlete. In brief, this included a series of obstacles (gymnastics vaulting boxes and benches) and open spaces set out within a gymnasium. The participants were required to navigate the course in the quickest way possible and were timed using timing gates positioned at the start and end points. Following two practice trials, each participant completed three trials with at least two-minutes between, and the best of the three trials was used in the analysis. A familiarisation session of the speed-run test was executed one week before the pre-intervention testing with data used against the pre-intervention measures to determine intra-class reliability (ICC) of the test.

**Training interventions.**   Participants of both experimental groups were required to complete a 15-minute warm-up once per week before their regular basketball practice across 8-weeks. The warm-up was led by the principal researcher (also a qualified S&C coach) and conducted in the same school gymnasium located in a separate building to the basketball practice. While one group completed their intervention, the other group completed low intensity shooting exercises with their basketball coach. This was portrayed to the players to relate to limited space available in the warm-up location. However, to account for any impact of the shooting exercises, the order by which each group completed the intervention (before or after shooting) was alternated each week. To ensure the time of the warm-ups was matched, a timer

was set for 15-minutes and commenced upon the explanation of the first activity / exercise of each of the respective warm-ups.

The details of the included exercises for both warm-ups are found in Table 1. For both groups, the coach adopted the same pedagogical approach, utilising a guided discovery strategy that provided limited technical instruction after the initial introduction to the movement skill / activity to be performed. This approach aligned to the typical practice of parkour coaches [34]. In addition to this, to prevent potential tedium in the Conventional Group, exercises were ordered differently in both groups across the 8-weeks. The control group, who were unaware of the warm-up interventions performed by the two experimental groups, instead continued with their normal basketball practice as well as other typical physical activities they were engaged in.

**Data analysis.** Within subject coefficient of variation (CV) and average CV measures for each test were determined using the spreadsheet software, Microsoft Excel (Microsoft Office 365). ICC calculation and inferential analyses were performed using the statistical analysis software, IBM SPSS Statistics for Windows, version 28.0. All measures were tested for normality using the Shapiro-Wilk test and for homogeneity using the Levene's test. To evaluate mean differences across the multiple variables, a Repeated Measures MANOVA was used to assess differences by group and time between pre- and post-testing for all three groups and across all measures except for the parkour speed-run. For the speed-run, a Repeated Measures ANOVA was used to assess differences by group and time.

In addition, due to the low dose application of the warm-up protocols, Cohen's $d$ was used to calculate within group effect sizes (ES) for each of the performance measures. The between group ES were also calculated to compare post-intervention measures between the two intervention groups. In both cases, the ES values were interpreted as 'small', 'medium', and 'large' in accordance with Cohen's guidelines [39]. For further practical understanding of the data, pre-post changes beyond within-subject coefficient of variation was also calculated for all measures.

## Phase 2 Qualitative data and analysis

**Semi-structured interviews.** Based upon recommendations by Ponizovsky-Bergelson et al. [40], qualitative interviews were conducted with eight of participants from the two experimental groups in Investigation 1 (four from the Conventional group and four from the Parkour group). Semi-structured interviews (Table 5) were used to illicit children's perspectives on the warm-up protocols. Each interview took place in the presence on a parent or guardian via the virtual meeting platform, Microsoft Teams (Redmond, Washington, USA), and was recorded

**Table 1. Exercises and activities included within the 15-minute warm-up for the respective experimental groups.**

| Conventional group (exercises from): | Parkour group (exercises from): |
|---|---|
| Body weight squatting | Tic-tac actions |
| Reverse lunge | Continuous bench vaults |
| Skipping for height / distance | Vault box jumps / mounts |
| Countermovement jumps | Vaulting |
| Drop landings (from toe raise) | Ground-based floor vaults |
| Accelerations (5–10 metres) | Leaping over benches (on to crash mats) |
| Ice skater jumps | Rope swings |
| Hip hinge (single and double leg) | |
| Short sprint races | |
| Hopping | |
| Push up variations | |

**Table 2. Average physical characteristics of the participants by group.**

| Group | Chronological age (years) | Height (cm) | Sitting height (cm) | Leg length (cm) | Mass (kg) | Maturity offset estimation (years) |
|---|---|---|---|---|---|---|
| Conventional | 10.96 ± 0.14 | 153.00 ± 7.54 | 75.50 ± 4.32 | 77.50 ± 4.85 | 42.45 ± 10.18 | -2.20 ± 0.93 |
| Parkour | 10.76 ± 0.23 | 148.80 ± 6.83 | 73.40 ± 2.70 | 75.40 ± 5.46 | 40.68 ± 6.74 | -2.76 ± 0.37 |
| Control | 11.96 ± 0.56 | 158 ± 3.99 | 80.42 ± 5.14 | 78.40 ± 5.54 | 45.53 ± 6.34 | -1.49 ± 0.73 |

for later transcription. All interviews lasted no more than 30-minutes in duration. Although it has been suggested that face-to-face interviews would have allowed for greater synchronous communication (e.g., social cues of the interviewee, such as body language) than virtual meetings [41,42], due to the COVID-19 pandemic a decision was made to use virtual rather than face to face meetings. Following each interview, the recording was transcribed using the transcription tool, Sonix (San Francisco, USA), after which, the transcriptions were checked for accuracy by the principal researcher.

**Data analysis.** A thematic analysis was undertaken using the codes developed through three rounds of iterative coding. In addition, inductive analysis technique were also utlised in the analysis of the transcripts, creating additional codes deemed to be pertinent to the study aims (see for e.g., Fereday and Muir-Cochrane, [43]). To code the data, similar to the methods used by Spaaij et al. [44], two of the investigators independently read each of the transcripts and text were coded against the preliminary codes using Excel (Author 1) and NVivo (Author 2). Initial meaning codes were then discussed and corroborated by both investigators before determining the axial coding scheme [44]. Following Ball et al., [45], Nvivo was used by Author 2 to digitally organize and manually code the data. The author did not use any of the advanced searching and coding functions to aid analysis.

## Results

Having been calculated to be approximately classified as either circa- or post-PHV, five participants were removed from the analysis. Additionally, due to low adherence levels (< 6 from a total of 9 exposures), a further three participants' data were removed from the analysis. In addition, one participant was removed due to injury. Therefore, a total of 18 participants who met the inclusion criteria relating to adherence, maturity status, and at least one year of participation in basketball were included in the statistical analyses (Table 2). In the analysis of the parkour-based speed run, only 10 participants were included. The descriptive data for all the participants is reported in Table 3.

A high degree of reliability was found between familiarisation scores and the pre-intervention test scores for the speed-run. Based on an absolute agreement, 2-way mixed-effects model, the ICC estimate was .963 with a 95% confidence interval from .600 to .994. The average CV for the familiarisation scores was 6.65%. Within subject variation (CV) values for all pre- and post-intervention tests are displayed in Table 4.

**Table 3. Descriptive pre- and post-intervention test measures.**

| Group | Pre- OHS knee flexion (˚) | Post- OHS knee flexion (˚) | ES | Pre-10-m time (s) | Post-10-m time (s) | ES | Pre-CMJ (cm) | Post-CMJ (cm) | ES | Pre-speed-run time (s) | Post-speed run time (s) | ES |
|---|---|---|---|---|---|---|---|---|---|---|---|---|
| Conventional | 119.16 ± 23.57 | 138.05 ± 27.67 | 0.71 | 2.12 ± 0.19 | 2.07 ± 0.15 | 0.35 | 21.66 ± 3.83 | 21.95 ± 3.54 | 0.14 | 9.17 ± 1.07 | 8.62 ± 0.92 | 0.56 |
| Parkour | 120.02 ± 30.02 | 117.59 ± 15.07 | -0.63 | 2.14 ± 0.10 | 2.16 ± 0.15 | -0.14 | 19.08 ± 4.97 | 19.15 ± 4.72 | 0.09 | 9.87 ± 1.42 | 9.39 ± 1.19 | 0.37 |
| Control | 138.21 ± 16.11 | 127.90 ± 29.64 | -1.12 | 1.96 ± 0.16 | 2.02 ± 0.18 | -0.37 | 22.82 ± 5.97 | 22.25 ± 4.80 | 0.11 | - | - | - |

Means, standard deviations (±) and within-group Cohen's *d* ES values are shown for each dependent measure.

**Table 4. Average pre- and post-intervention coefficient of variation (%) per group.**

| | OHS Pre-knee flexion angle left | OHS Post-knee flexion angle left | OHS Pre-knee flexion angle right | OHS Post-knee flexion angle right | Pre-10-m | Post-10-m | Pre-CMJ | Post-CMJ | Pre-speed-run time | Post-speed run time |
|---|---|---|---|---|---|---|---|---|---|---|
| Conventional Group | 2.49 | 3.34 | 1.96 | 3.24 | 2.82 | 2.05 | 5.86 | 3.70 | 3.58 | 2.85 |
| Parkour Group | 1.62 | 3.91 | 1.84 | 3.26 | 2.50 | 1.51 | 3.89 | 4.82 | 2.82 | 1.59 |
| Control Group | 3.30 | 1.85 | 3.50 | 3.30 | 0.95 | 2.08 | 4.32 | 4.04 | - | - |

All pre- and post-intervention data was determined to be normally distributed ($p > 0.5$). The Repeated Measures MANOVA revealed no significant effects of group on pre-post intervention measures, $F_{(6, 22)} = .793^b$, $p > .05$, Wilk's $\Lambda = .676$, partial $\eta^2 = .178$. In addition, no significant between subjects effects were observed for time, $F_{(3, 11)} = .092^b$, $p > .05$, Wilk's $\Lambda = .975$, partial $\eta^2 = .025$. Following this, using partial $\eta^2$ to determine effect size, a post-hoc power analysis for between subjects' effects revealed effect size F = 0.312 and statistical power (1- β err prob) to be 0.25.

The repeated measures ANOVA used for the analysis of the parkour-based speed run revealed no significant effects of time x group interaction on completion times, $F_{(1, 9)} = .219^b$, $p > .05$, Wilk's $\Lambda = .976$, partial $\eta^2 = 0.24$. A post-hoc power analysis revealed the effect size F = 0.562 and statistical power (1- β err prob) to be 0.91.

In the Conventional warm-up group, the within group ES values revealed a medium ES improvement in knee flexion angle in performance of the OHS. In contrast, the Parkour and Control groups displayed reductions in knee flexion angles with a medium and large ES, respectively. For the Conventional warm-up group, the Cohen's *d* yielded a small ES for the 10-m sprint. In contrast, the Parkour and Control groups displayed increases in 10-m sprint times with small ES though the magnitude of the increase was greater in the Control group. For the CMJ, across each group, the within group ES was found to be small. In the speed-run, a medium ES was revealed for the Conventional group, while the Parkour warm-up group displayed a small ES for their pre-post speed-run times. The between-group ES values for the two experimental groups were 0.61 for the 10-m sprint, 0.73 for the speed-run, 0.62 for the CMJ, and 0.98 for the OHS, representing medium to large effects across all measures.

Figs 1–4 provide individual pre- and post-intervention data across each of the performance measures. Dashed lines have been used to represent individuals showed percentage changes greater than pre-intervention CV, while solid lines have been used to indicate that changes were less than pre-intervention CV.

## Qualitative findings

Data was categorised into three higher order themes drawing on data from the young players' responses: enjoyment; physical literacy; and docility. These themes included subthemes that related to the young players' reflections on the value and purpose of the warm-up intervention and perceived benefits on basketball playing performance (see Table 5).

**Theme 1 –*Enjoyment*.** Most participants indicated that they enjoyed the warm-up activities, irrespective of the experimental group they were assigned to:

"Yeah, it's definitely one of the things that I enjoy doing a lot because it's not just like running there and back, but it's including like jumps and then like moving around more rather than going in the straight line there and back."

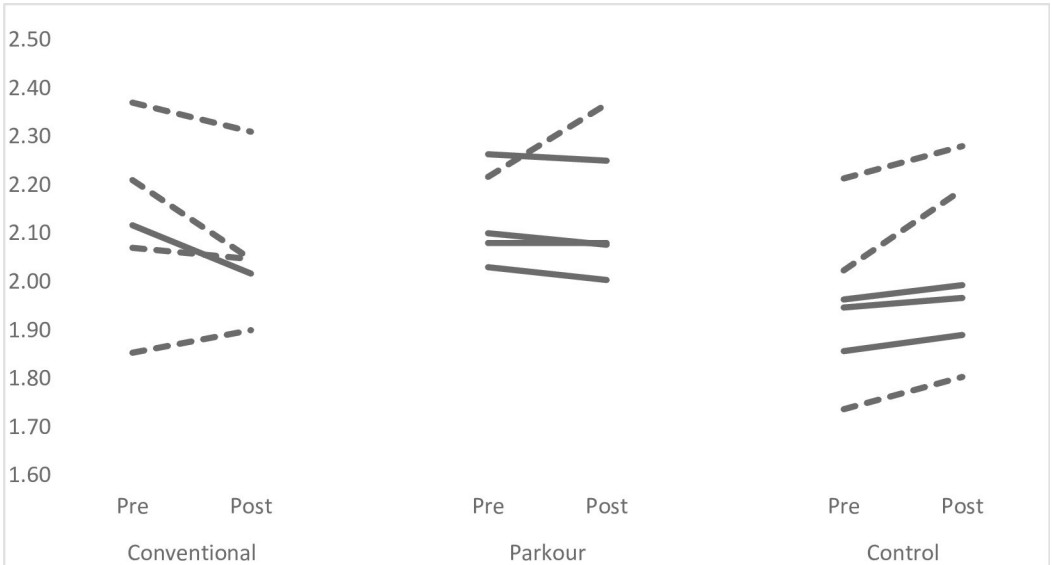

**Fig 1. Individual pre-post intervention mean 10-m sprint data.** Dashed lines represent % changes > than pre-intervention CV; solid lines represent difference that was not > CV.

With specific reference to parkour-based activities, participants also suggested that they found the warm-up to be fun. One individual commented:

"I think honestly, I really like jumping over the things because I found it fun. It was like when and also jumping onto the mat. That was quite fun as well. And obviously the ropes at the end. That was just the fun."

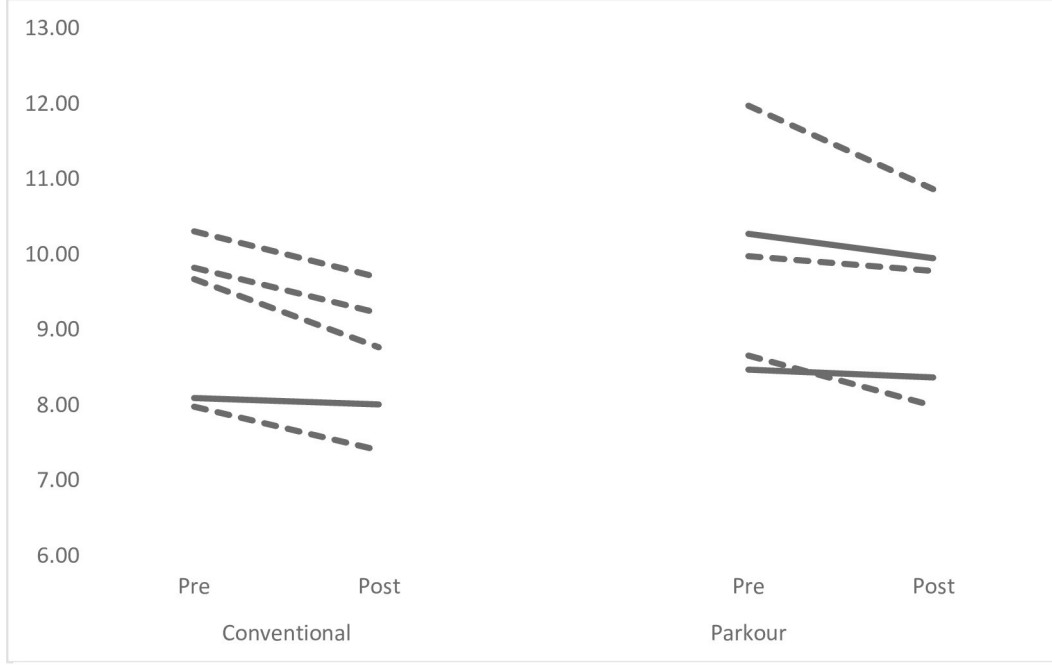

**Fig 2. Individual pre-post intervention mean Speed-Run data.** Dashed lines represent % changes > than pre-intervention CV; solid lines represent difference that was not > CV.

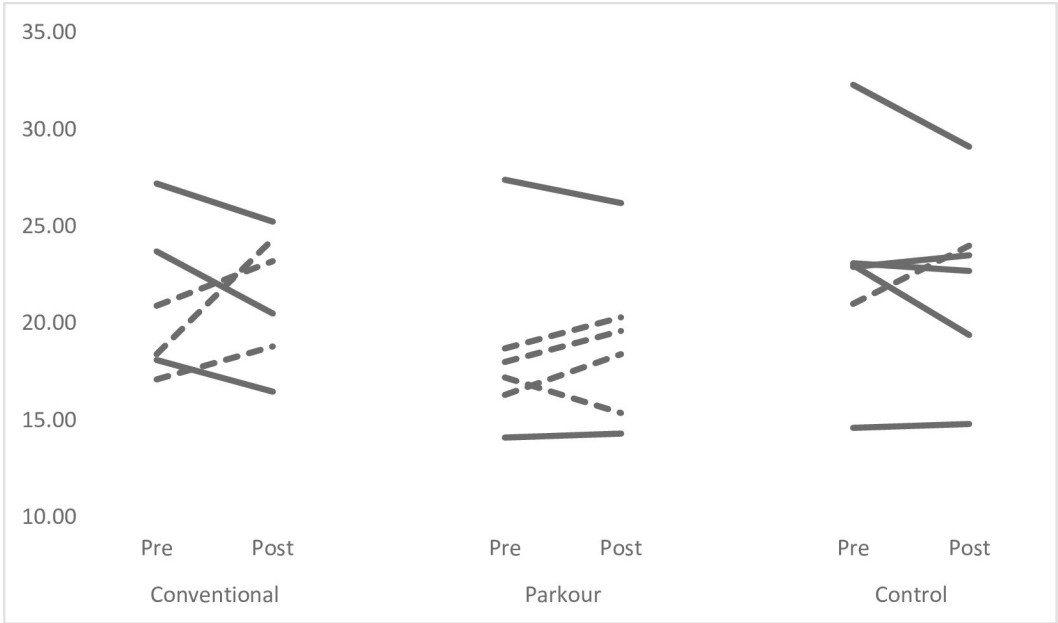

**Fig 3. Individual pre-post intervention mean CMJ data.** Dashed lines represent % changes > than pre-intervention CV; solid lines represent difference that was not > CV.

Similarly, another individual stated:

"Yeah, because it was it was [sic] a good time doing it. It was a good part of the day, like. Oh year, it's fun."

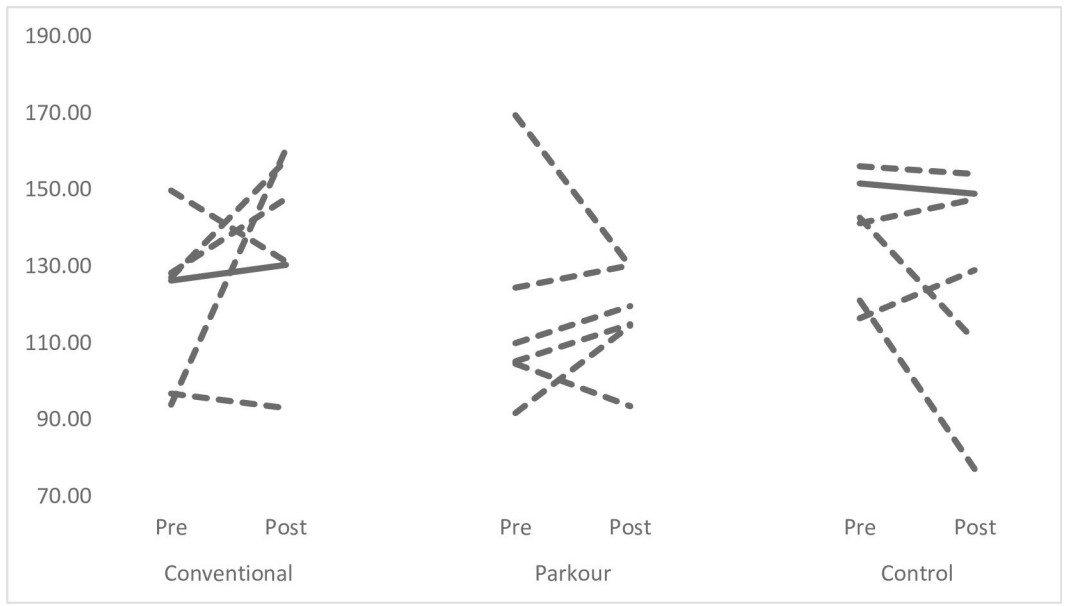

**Fig 4. Individual pre-post intervention mean Overhead Squat knee flexion data.** Dashed lines represent % changes > than pre-intervention CV; solid lines represent difference that was not > C.

**Table 5. Higher order themes and associated subthemes.**

| Docility | Enjoyment | Physical Literacy |
|---|---|---|
| despair | free play | autonomy |
| lack of enjoyment | fun | confident |
| performativity | Getting ready | critical (of activities prescribed) |
| parent | improvement | inquiry |
| | new | reflection |
| | | self-awareness |
| | | specificity |

**Theme 2 –*Physical Literacy*.**    Improved confidence in relation to movement competency and motor abilities was identified by the participants. In addition, participants displayed critical reflection of the activities prescribed and self-awareness of their movement capabilities. When asked whether the warm-up activities benefitted basketball, one participant reflected:

> "The rope swing? Yeah, I think those might be less applicable to basketball, but they still help upper body strength."

Another participant responded with:

> "For me, I think it was like during the sprint. Uh, because that was the bit that helped me the most. And also because because [sic] like it, it was a bit more competitive than most of the other warm ups we did."

And, another participant stated:

> "It helped me like [sic] control my speed levels. . .I can like [sic] fake it"

**Theme 3 –*Docility*.**    In some of the participants, docility was detected through responses that conveyed an indifferent attitude or appeared to indicate a level of performativity. In response to whether they enjoyed the warm-up activities one individual commented:

> "I don't know. I can't think right now that nothing was not fun. I liked it."

Another individual stated:

> "Like the obstacle course, all the stuff we did. And, yeah."

## Discussion

The results of the quantitative phase of our study revealed no differences between conventional neuromuscular training exercises and parkour-based actions when utilised within the warm-up protocols of pre-PHV basketball players. In relation to the Conventional warm-up group, our findings appear to contradict previous studies that have highlighted the efficacy of NMT-based warm-up programmes [22,25,28,46]. Where typically the purpose of NMT-based warm-

ups is to improve neuromuscular control and force related outputs [22], the warm-up interventions used in our study did not appear to elicit these particular adaptations. Although some likely explanation for our results is related to the small sample size, due to the high-levels of neural plasticity in pre-PHV populations, athletic development models typically advocate for breadth of movement skills development ahead of the adolescent growth spurt, after which force characteristics may be more readily enhanced [10,28]. Thus, where the emphasis is on breadth of movement skills, the corollary may be limited display of improved athletic capabilities (e.g., CMJ and 10-m sprint speed). Another possible explanation for our results is that the stimulus of the respective warm-ups may not have been sufficient to enhance the physical capabilities of the young players [25]. In this regard, the adopted pedagogical approach may have limited the consistency of stimulus exposure across both warm-up interventions. The guided discovery approach, along with the decision to match the workload by time to better accommodate the parkour-based content, is likely to have reduced the total volume of work in each of the exercises, and, in turn, the magnitude of the stimulus. Guided discovery enables the learner to explore multiple possibilities rather than following a traditional coach-led and narrow technique-based pedagogy [47,48]. However, inherently, this method may require more time for improvements to occur compared to the coach-led approach using direct instruction [49]. Given the lack of training experience and exposure to NMT-based warm-ups by the participants in our study, it is possible that the guided discovery approach limited the development of the skills and athletic capabilities in the two experimental groups. Therefore, though this approach is considered beneficial for long-term development and skill transfer to other activities [48,50], where the aim of the NMT programme is to improve movement skills and enhance general physical qualities, this pedagogical approach may not be optimal for short term development of athletic capabilities. Nonetheless, this delivery approach would appear to align to the wider aims of PL by enabling the young individuals to explore movements and perform skills without the constraints of strict technical models [51].

In addition to the pedagogical delivery strategy, another possible explanation for our results may relate to the frequency of the warm-up exposure being limited to once per week. Typically, studies that have highlighted the efficacy of NMT-based warm-ups have prescribed two exposures per week (e.g., [46,52,53]). Similarly, the length of the intervention period may have been a contributing factor to our results. The meta-analysis by Faude et al. [22], which observed the effects of NMT injury prevention programmes, found larger effects with >23 training sessions compared to < 23. However, other studies (e.g., [25,46] have observed improved performance in response to only 4-weeks exposure to NMT-based warm-up protocols. However, these studies have exposed participants to three sessions per week, thus reaffirming the importance of frequency. Therefore, the efficacy of relatively short NMT-based warm-up intervention periods may be dependent upon the frequency of exposures. It is likely, therefore, that the single weekly exposure across an 8-week period in our study was not sufficient to lead to significant changes.

Notwithstanding the results of our multivariate analyses, comparisons of within-group ES values for pre-post measures appeared to demonstrate that some specific adaptations were elicited in response to the stimuli of the respective warm-up programmes. Specifically, there were observed improvements in speed-related measures for the Conventional group, whereas both the Parkour and control groups showed a tendency to worsen in 10-m sprint performance, the largest effect of which was found in the control group. In the parkour-based speed-run test, however, while the largest within-group effect size was found for the Conventional group, the Parkour group also showed improvements in performance. In contrast, pre-post measures for the CMJ did not reveal any distinct changes, with very small effect sizes across all three groups. In the OHS knee flexion angles were found to improve with moderate effect sizes in the

Conventional group and worsen in the Parkour and control groups. However, as might be expected, irrespective of group, at an individual level, each of the test measures revealed mixed results with some participants appearing to either show positive or negative changes greater than their pre-intervention CV. Thus, despite good levels of pre- and post-test reliability, the observed effect sizes were somewhat influenced by outliers, with some participants displaying substantially large differences between pre- and post-intervention measures, perhaps highlighting challenges associated with physical testing for empirical studies in preadolescent youths. In this regard it has previously been highlighted that in younger athletic populations, there may be increased variability in test performance due to limited physical development [54]. Notably, however, in accordance with the results for within-group ES, most participants that improved their test measures beyond their pre-intervention CV were in the two experimental groups. Importantly, therefore, versus the control group, the two warm-up interventions appeared to provide stimuli that to some extent preserved physical fitness qualities, despite being characterised by low volumes and frequencies. This is of relevance to youth athletic populations who, through specialisation in a single sport, have been highlighted as being at risk of underdevelopment of movement capabilities and physical development of general fitness qualities [55,56] and, in turn, may be at may greater risk of injury [57–59].

Despite both warm-up interventions potentially preserving the young athletes' physical fitness qualities, the medium to large between group ES, taken with the within-group ES difference across groups, appear to suggest that, for certain qualities, the Conventional warm-up was more effective than the Parkour-based warm-up. For example, the Conventional group's exposure to acceleration speed is a likely explanation for their improvement in measure for 10-m sprint speed compared to the other groups. This may also account for the observed differences in OHS knee flexion achieved by the Conventional group compared to the Parkour and control groups. While the OHS was purposefully not included in the Conventional group's warm-up, a bodyweight squat pattern (with arms held in front of the body) was included in each training session. Where the ability to perform a bodyweight squat to a depth of at least 90˚ of knee flexion (or thighs parallel to the floor) is considered an indication of movement quality and neuromuscular control and movement skill [60,61], it is likely that exposing the participants to various squat patterns contributed to improvements in the OHS.

Somewhat contradicting to the apparent specific responses to the respective warm-up interventions' content is observed in the Parkour group's small ES value for the CMJ, despite being characterised by a greater volume of jumping and leaping activities compared to the Conventional group. However, as indicated by the observed changes across the groups against the pre-intervention CV values, there appeared to somewhat similar patterns of improvement in response to the different two warm-up interventions. A plausible explanation, therefore, is that the lack of prescription of exercise repetitions / foot contacts in the Parkour group in comparison to typical NMT-based warm-up programmes that prescribe progressively increasing volumes for each exercise (e.g., FIFA-11+). Indeed, the comparative results for the Conventional group, suggest that the low-structured prescription of exercises that characterised both interventions may have limited the development of jumping-related qualities.

In summary, our quantitative results suggest that 15-minute NMT-based warm-up interventions offer some preservative benefits to the physical fitness qualities of pre-PHV athlete group. Moreover, as indicated in the results of the Parkour group's warm-up intervention versus the control, there is potential merit to the incorporation of less conventional activities and exercises with the youth athletic development strategy.

## Qualitative research better psycho-social and embodied outcomes consistent with phenomenological definitions of PL

The thematic analysis revealed that the intervention warm-ups may have aligned to the concept of holistic development of the young basketball players that is typically emphasised within youth athletic development literature [11,12,36]. Such an approach goes beyond physical training outcomes (e.g., muscular fitness and strength) and includes cognitive training, such as social interaction and stress management [31]. Accordingly, it has been suggested that for children, the focus of training should be placed on fun-based activities that are geared towards preparatory conditioning [31,62]. Both of the intervention warm-ups used in the current study appeared to create a sense of enjoyment in the participants, with multiple references to fun made by the interviewed children. This may have related to the pedagogical delivery of the warm-up activities, though it is possible that the novelty of the movement patterns and actions resulted in the feelings of fun experienced by the participants. Of importance, responses from the children that highlighted the notion of fun and enjoyment appeared to be specifically related to parkour-based activities. This was indicated by responses from the Conventional group that appeared to relate to the speed run test, which required participants to navigate and overcome various obstacles. Therefore, while no significant differences were observed in the physical performance measures between the two experimental groups, the parkour activities might be a more effective means to create engagement through increased levels of enjoyment.

While the Parkour warm-up group indicated greater levels of enjoyment, both groups appeared to display self-reflection and critical thought in relation to the included activities. In this regard, the responses contributed to the theme *physical literacy (PL)*, which, as a concept, relates to the confidence and physical competence, as well as the knowledge and understanding, to engage in physical activity across the lifespan [4,5,63]. Indeed, confidence appeared as a subtheme within the higher order PL theme. Across both warm-up groups, participants referred to feeling a sense of increased self-confidence. Some individuals referred to specific aspects of their game, for example a participant from the Conventional group stated:

> "*I've been more confident in the things that I know . . .I'm able to do some of these things now rather than before. But now I know I'm able to do it so I can definitely give it a go*".

The above example highlights the wider implications of the warm-up that extend to psychological-based outcomes. In relation to this, PL extends beyond physical capacities and encompasses perception, memory, experience and decision-making [4]. Indeed, the display of self-reflection in the responses of the young basketball players, for example the comments relating the rope swing's relevance to basketball and another participant's reference to sprint races that "*helped the most*" suggests that they perceived benefits to their own performance capabilities. Such reflection and search for meaning can be considered to relate to the philosophical underpinnings of PL, including existentialism and phenomenology [64,65]. These underpinnings are closely aligned and relate to an individual's experiences and perceptions of the world around them and the meaning that the individual derives [64]. In this regard, the ability of the young basketball player to think about and contextualise the relevance of the warm-up activities to basketball performance highlights the occurrence of learning through movement, which is representative of the holistic nature of PL [66]. Potentially, the novelty and explorative nature of the activities, as well as the pedagogical delivery approach, may have provoked the young basketball players to contextualise the meaning of the warm-up. However, what is not known is whether the young players would demonstrate this same level of reflection to other warm-up activities not delivered with the same pedagogical approach.

The third and final theme, *docility*, appears to contradict the notion of reflective and critical thinking. A possible explanation for this, however, may have been related to the nature of semi-structured interviews. The online medium used to conduct the interviews may have influenced the young players, causing them to appear docile in response to the questions. While online interviews have been suggested to be as effective as in-person interviews [41], interviews in person have been found to result in more words spoken by adolescent respondents [67]. In this regard, the docility may well have been temporary rather being associated to any deeper meaning. Further possible explanation might relate to the open questions and the interviewer attempting to avoid leading questions and biasing the responses given by the young interviewees. The challenge of interviewing children has been previously discussed by Ponizovsky-Bergleson [40], who suggests that children have a tendency to respond to questions in an obligatory manner. Indeed, within the docility theme, the subtheme, *performivity*, was also identified. In this regard, children appeared to provide answers that they felt the interviewer, or indeed their parents, wished to hear. Strategies to reduce performivity include question request (e.g., "can you explain that to me?"), and encouragement (e.g., statements of approval) [40]. However, it is possible that in online interviews where body language is difficult to determine [67], as well as the need for parental guidance, there may trade-off between docility and performivity in the participants.

## Conclusions

Collectively, the results of our two investigations suggest that NMT-based warm-ups can be effective in the broader development of pre-adolescent basketball players beyond the typical aims and objectives of athletic development. Although limited, our findings highlight potential benefits of parkour-related activities alongside typical neuromuscular-focused exercises, as part of the youth athletic development strategy. Despite low frequency of exposure, the incorporation of less conventional approaches, such as the use of parkour-related activities within a low structured learning environment, may at the very least, preserve athletic capabilities ahead of PHV. Moreover, from a holistic perspective, this approach would appear to contribute to broader aims of PL, including the development of FMS and qualities of physical fitness (e.g., speed, strength, jumping ability), critical reflection, and self-confidence, while evoking a combined sense of enjoyment, fun, and purpose.

## Supporting information

**S1 Data.**
(XLSX)

## Author Contributions

**Conceptualization:** Mark David Williams.

**Data curation:** Mark David Williams, Andrew Hammond.

**Formal analysis:** Mark David Williams, Andrew Hammond.

**Methodology:** Mark David Williams, Jason Moran.

**Project administration:** Mark David Williams.

**Supervision:** Jason Moran.

**Writing – original draft:** Mark David Williams.

**Writing – review & editing:** Mark David Williams, Andrew Hammond, Jason Moran.

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
