## [Decision Letter · Decision Letter 0]

13 Sep 2022

PONE-D-22-19812Beyond athletic development: the effects of parkour-based versus conventional neuromuscular exercises in pre-adolescent basketball playersPLOS ONE

Dear Dr. Williams,

Thank you for submitting your manuscript to PLOS ONE. After careful consideration, we feel that it has merit but does not fully meet PLOS ONE’s publication criteria as it currently stands. Therefore, we invite you to submit a revised version of the manuscript that addresses the points raised during the review process.

ACADEMIC EDITOR:Dear authors,

reply point by point to the reviewer's comments.

The manuscript needs major revision to improve.

Follow the reviewers' suggestions carefully.

We look forward to receiving your revised manuscript.

Kind regards,

Gianpiero Greco

Academic Editor

PLOS ONE

Journal Requirements:

Reviewers' comments:

Reviewer's Responses to Questions

**Comments to the Author**

1. Is the manuscript technically sound, and do the data support the conclusions?

Reviewer #1: Yes

Reviewer #2: Partly

2. Has the statistical analysis been performed appropriately and rigorously? 

Reviewer #1: Yes

Reviewer #2: I Don't Know

3. Have the authors made all data underlying the findings in their manuscript fully available?

Reviewer #1: Yes

Reviewer #2: No

4. Is the manuscript presented in an intelligible fashion and written in standard English?

Reviewer #1: Yes

Reviewer #2: No

5. Review Comments to the Author

Reviewer #1: It is a pleasure to review this manuscript.

It is a well written article. It brings a lot of novelty in sport sciences.

I would congratulate the authors for the works' quality.

I just have minor suggestions: Please could you add the intraclass correlation coefficient (ICC) for the used test and the used type of the ICC.

I would appreciate that authors add a table describing the characteristics of the participants (body mass, height, sitting height, maturity off set and APHV for each group). It would be nice if the authors add this characteristics before and after the intervention period.

Reviewer #2: The proposed manuscript seems interesting and has a noteworthy objective. Mixing quantitative with qualitative analysis was also interesting, however, lack of significant between-group differences does not help. The fact that the intervention was only held once a week, may have not been enough to observe change. Can it be a pilot study? Did the authors try to analyze the data with a General Linear Model Univariate procedure to test for any significant changes that may support the manuscript? Unfortunately, in my opinion, as is, the manuscript does not meet criteria to be recommended for publishing in this journal.

6. PLOS authors have the option to publish the peer review history of their article (what does this mean?). If published, this will include your full peer review and any attached files.

Reviewer #1: **Yes: **Yassine Negra

Reviewer #2: No

---

## [Author Response · Author response to Decision Letter 0]

10 Oct 2022

Thank you for taking the time to review our manuscript and provide us with valuable feedback. All comments have been reviewed and addressed within our response to reviewers document. We thank you again for considering our manuscript for publication in PLOS One.

---

## [Decision Letter · Decision Letter 1]

13 Dec 2022

PONE-D-22-19812R1

Beyond athletic development: the effects of parkour-based versus conventional neuromuscular exercises in pre-adolescent basketball players

PLOS ONE

Dear Dr. Williams,

Thank you for submitting your manuscript to PLOS ONE. After careful consideration, we have decided that your manuscript does not meet our criteria for publication and must therefore be rejected.

Specifically:

ACADEMIC EDITOR:

In my opinion, in present form, the manuscript does not meet criteria to be recommended for publishing in this journal. 

The manuscript needs extensive review before it can be resubmitted to this or any other journal.

I am sorry that we cannot be more positive on this occasion, but hope that you appreciate the reasons for this decision.

Kind regards,

Gianpiero Greco

Academic Editor

PLOS ONE

Reviewers' comments:

Reviewer's Responses to Questions

**Comments to the Author**

1. If the authors have adequately addressed your comments raised in a previous round of review and you feel that this manuscript is now acceptable for publication, you may indicate that here to bypass the “Comments to the Author” section, enter your conflict of interest statement in the “Confidential to Editor” section, and submit your "Accept" recommendation.

Reviewer #2: All comments have been addressed

Reviewer #3: (No Response)

2. Is the manuscript technically sound, and do the data support the conclusions?

Reviewer #2: Partly

Reviewer #3: No

3. Has the statistical analysis been performed appropriately and rigorously? 

Reviewer #2: I Don't Know

Reviewer #3: No

4. Have the authors made all data underlying the findings in their manuscript fully available?

Reviewer #2: Yes

Reviewer #3: No

5. Is the manuscript presented in an intelligible fashion and written in standard English?

Reviewer #2: Yes

Reviewer #3: Yes

6. Review Comments to the Author

Reviewer #2: Thank you for addressing my comments.

Please reconsider the conclusions reached withou the necessary significance between groups.

These results cannot be extrapolated to the participants of this sport. Milder conclusions are needed to meet the lack of significant results.

Reviewer #3: GENERAL COMMENTS

Overall, this is a good quality, clear, and well-written manuscript. The authors have collected noteworthy dataset and made a systematic contribution to the research literature in this area of investigation, with a strong potential for practical applications. I would like to thank the authors for their efforts in working towards improving sports’ science knowledge. I praise you for the initiative and efforts. Nevertheless, there is an area of shadow and some clarifications that have to be made in relation to the study’s statistical tool.

The major issue with this manuscript is linked with the use of Magnitude-based inference (MBI). The author used the Hopkins’ scale (Hokins et al 2009) to compare the parkour-based, the conventional and the control groups. MBI is a controversial statistical method that has been used in hundreds of papers in sports science despite criticism from statisticians. What about the guidelines highlighted by Sainani (2018)? Analysis using the smallest worthwhile change should no more be tolerated in future sport’s science publications.

References

Sainani KL. The problem with “magnitude based inference”. Med Sci Sports Exerc. 2018;50 (10):2166-76.

7. PLOS authors have the option to publish the peer review history of their article (what does this mean?). If published, this will include your full peer review and any attached files.

Reviewer #2: No

Reviewer #3: **Yes: **Mohamed Souhaiel Chelly

- - - - -

---

## [Author Response · Author response to Decision Letter 1]

26 Feb 2023

We thank the reviewers for their time and effort given to reviewing our manuscript and value their comments. We have addressed the areas of concern and have made major revisions in accordance with these. 

We have submitted a file that addresses each of the comments given.

---

## [Decision Letter · Decision Letter 2]

28 Jun 2023

Beyond athletic development: the effects of parkour-based versus conventional neuromuscular exercises in pre-adolescent basketball players

PONE-D-22-19812R2

Dear Dr. Williams,

We’re pleased to inform you that your manuscript has been judged scientifically suitable for publication and will be formally accepted for publication once it meets all outstanding technical requirements.

Kind regards,

Emiliano Cè

Academic Editor

PLOS ONE

Additional Editor Comments (optional):

Reviewers' comments:

Reviewer's Responses to Questions

**Comments to the Author**

1. If the authors have adequately addressed your comments raised in a previous round of review and you feel that this manuscript is now acceptable for publication, you may indicate that here to bypass the “Comments to the Author” section, enter your conflict of interest statement in the “Confidential to Editor” section, and submit your "Accept" recommendation.

Reviewer #1: All comments have been addressed

Reviewer #4: All comments have been addressed

2. Is the manuscript technically sound, and do the data support the conclusions?

Reviewer #1: Yes

Reviewer #4: Yes

3. Has the statistical analysis been performed appropriately and rigorously? 

Reviewer #1: (No Response)

Reviewer #4: Yes

4. Have the authors made all data underlying the findings in their manuscript fully available?

Reviewer #1: Yes

Reviewer #4: Yes

5. Is the manuscript presented in an intelligible fashion and written in standard English?

Reviewer #1: Yes

Reviewer #4: Yes

6. Review Comments to the Author

Reviewer #1: (No Response)

Reviewer #4: (No Response)

7. PLOS authors have the option to publish the peer review history of their article (what does this mean?). If published, this will include your full peer review and any attached files.

Reviewer #1: **Yes:**

Reviewer #4: **Yes:**

---

## [Editor Report · Acceptance letter]

4 Jul 2023

PONE-D-22-19812R2 

Beyond athletic development: the effects of parkour-based versus conventional neuromuscular exercises in pre-adolescent basketball players 

Dear Dr. Williams:

I'm pleased to inform you that your manuscript has been deemed suitable for publication in PLOS ONE. Congratulations! Your manuscript is now with our production department. 

Kind regards, 

on behalf of

Prof. Emiliano Cè 

Academic Editor

PLOS ONE